# Probabilistic controllability approach to metabolic fluxes in normal and cancer tissues

Jean-Marc Schwartz [1], Hiroaki Otokuni[2], Tatsuya Akutsu [3] & Jose C. Nacher [2]

Recent research has shown that many types of cancers take control of specific metabolic processes. We compiled metabolic networks corresponding to four healthy and cancer tissues, and analysed the healthy–cancer transition from the metabolic flux change perspective. We used a Probabilistic Minimum Dominating Set (PMDS) model, which identifies a minimum set of nodes that act as driver nodes and control the entire network. The combination of control theory with flux correlation analysis shows that flux correlations substantially increase in cancer states of breast, kidney and urothelial tissues, but not in lung. No change in the network topology between healthy and cancer networks was observed, but PMDS analysis shows that cancer states require fewer controllers than their corresponding healthy states. These results indicate that cancer metabolism is characterised by more streamlined flux distributions, which may be focused towards a reduced set of objectives and controlled by fewer regulatory elements.

[1] Faculty of Biology, Medicine and Health, University of Manchester, Manchester M13 9PT, UK. [2] Department of Information Science, Faculty of Science, Toho University, Funabashi 274-8510, Japan. [3] Bioinformatics Center, Institute for Chemical Research, Kyoto University, Uji 611-0011, Japan. Correspondence and requests for materials should be addressed to J.-M.S. (email: jean-marc.schwartz@manchester.ac.uk) or to J.C.N. (email: nacher@is.sci.toho-u.ac.jp)

Metabolic pathways are essential chemical processes that catalyse complex reactions indispensable for development and life. Recent research has shown that many types of cancers take control of specific metabolic processes[1]. In particular, cancer pathogenesis has recently been closely associated to serine, glycine and one-carbon metabolism[2]. Preclinical analyses have shown that by adopting a specific diet that limits the amount of serine and glycine, tumour growth was severely inhibited.

Understanding how metabolic networks are controlled is a challenging question though. Metabolic control analysis was originally developed to evaluate how metabolic fluxes depend on kinetic parameters of enzymatic reactions or metabolite concentrations. The level of control is quantified by calculating control or elasticity coefficients[3,4]. However, these approaches are not easily applicable to large systems since they rely on a kinetic model of the metabolic pathways. More recently, an approach was developed that uses flux couplings between metabolic reactions[5]. Flux couplings identify constrained relations between fluxes of different reactions, which arise from the network topology and mass conservation[6]. Five types of couplings were identified, namely directional, partial, full, anti and inhibitive couplings. The networks were analysed using a minimum dominating set (MDS) approach to identify potential driver nodes reactions[5]. The MDS approach was proposed by Nacher and Akutsu[7] and has been used to analyse many biological systems and identify proteins associated to cancer[8–12]. However, the flux coupling approach uses a discrete definition of coupling between reactions and does not take into account the full range of possible flux distributions that the metabolic system is able to support. For example, two reactions are defined as anti-coupled if when one of them is inactive then the other carries a non-zero flux (in a non-zero steady state), but this definition does not take into account the relation between both fluxes in situations where both reactions are active.

Here, we argue that in order to investigate the distributed control of metabolic flux in large-scale metabolic networks, a different approach is needed. Coupling should be measured by a continuous value characterising the relations between reaction fluxes over all feasible steady states, instead of a set of binary values that represent particular subsets of steady states. In order to distinguish this continuous measure from previously used definitions of coupling, we hereon use the term correlation. Such a measure already exists: as demonstrated by Poolman et al.[13], it can be obtained from the angles between the vectors forming an orthonormal basis of the null-space (or kernel) of the stoichiometric matrix defining the metabolic network. The cosine of this angle precisely represents Pearson's correlation coefficient between the fluxes carried by a pair of reactions over all possible steady states of the system, and was therefore named reaction correlation coefficient. Hence, the reaction correlation coefficient $\phi_{ij}$ of a pair of reactions $i$ and $j$ is a continuous value comprised between –1 and 1, where $\phi_{ij} = 0$ indicates that the fluxes of both reactions are completely independent, $\phi_{ij} = 1$ indicates that they are perfectly correlated and $\phi_{ij} = -1$ indicates that they are perfectly anti-correlated.

In this work, we compiled metabolic networks corresponding to four healthy and cancer tissues, namely breast, lung, kidney and urothelial cancer. This data allows us to analyse the transition from healthy to cancer states from the metabolic flux perspective (Fig. 1). We first assembled a metabolic flux correlation network obtained from biochemical metabolic pathways in healthy and cancer tissues. Indeed, the above-mentioned flux correlation coefficient can be interpreted as a failure probability, which suggests that a probabilistic model can be suitable to address flux control. For example, a positive correlation interaction with value 1 could be understood as an interaction with failure probability of zero. Similarly, an absence of correlation with value zero would suggest a failure probability of 1. These concepts pave the way to the applicability of a probabilistic control theory for complex metabolic networks. Here, we used a Probabilistic Minimum Dominating Set (PMDS) model that can identify a minimum set of nodes that act as driver nodes and control the entire network in a context of probabilistic

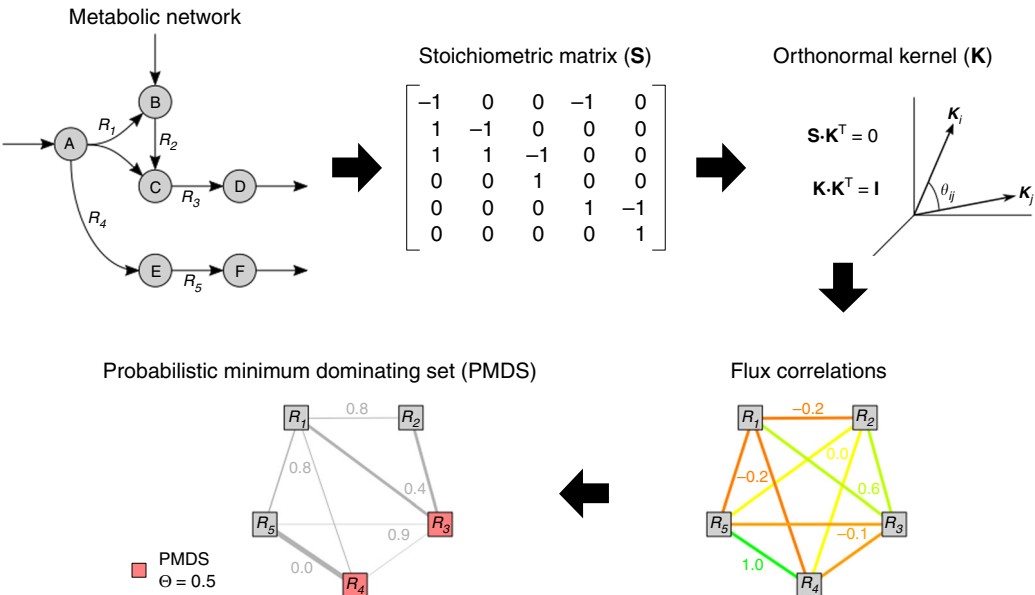

**Fig. 1** Method for the application of probabilistic control theory to metabolic flux analysis. The stoichiometric matrix of the metabolic network is constructed and used to compute the orthonormal kernel matrix. The reaction correlation coefficient $\phi_{ij}$ is the cosine of the angle between the reaction rows in the kernel and represents the strength of control between both reactions. The failure probability $\rho_{ij}$ is defined as $1-|\phi_{ij}|$, which is used to determine the probabilistic minimum dominating set. We require that each node (reaction) is covered by multiple nodes in PMDS so that the probability that at least one edge (incoming flux) is active is greater than the threshold $\Theta$

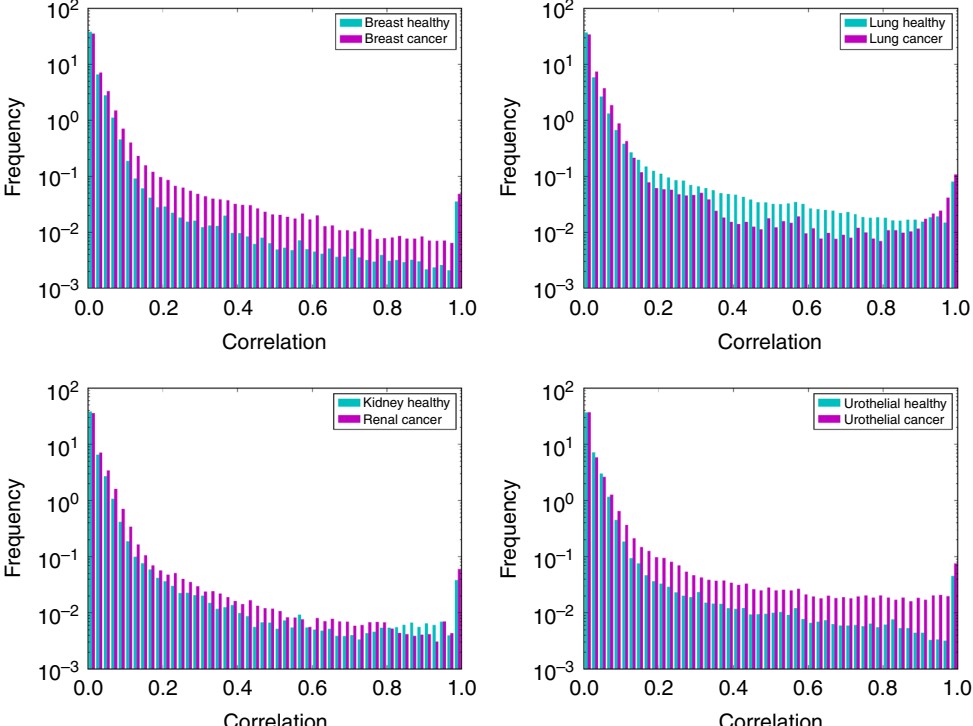

**Fig. 2** Probability distribution of reaction correlations. Probability distribution of reaction correlations in healthy (blue) and cancer (purple) states of four human tissues

interaction failures[14]. In this network representation, the nodes are interpreted as metabolic reactions and the weighted edges correspond to the probabilistic flux exchanged among reactions. The application of PMDS in the context of metabolic fluxes makes it possible to connect controllability theory with flux correlations.

## Results

**Cancer metabolic reaction fluxes show higher correlations.** Frequency distributions of all correlations in models of healthy and cancer cells corresponding to four different tissues were computed and the results are shown in Fig. 2. In general, we observed that the correlations are stronger in the cancer state than the healthy state for breast, kidney and urothelial tissues. However, for lung cancer the results show an opposite tendency with a weaker correlation distribution for the cancer state.

**Statistical analysis of genome-scale metabolic flux networks.** The clustering degree in all networks shows, however, a similar local structure (Table 1). This indicates that the differences observed in cancer states are not due to changes in the network structure, but to changes in the relations between fluxes.

We also determined the degree distributions in all these networks and found that they are similar and adhere to the classical scale-free distribution observed in most biological networks, including metabolic networks (Fig. 3). It is interesting that in spite of large metabolic flux changes, the global statistical pattern of the network remains conserved.

However, the specificity of lung cancer appears in the fraction of nodes connected to the giant connected component (Table 2 and Fig. 4). Breast, urothelial and renal cancers show larger giant components than the corresponding healthy states. In contrast, healthy lung tissue shows a larger giant component than the

**Table 1 Topological properties of the complete metabolic networks**

| Network | Clustering coefficient | Connected components |
|---|---|---|
| Breast healthy | 0.780 | 4 |
| Breast cancer | 0.766 | 7 |
| Lung healthy | 0.772 | 5 |
| Lung cancer | 0.771 | 8 |
| Kidney healthy | 0.770 | 6 |
| Renal cancer | 0.764 | 5 |
| Urothelial healthy | 0.780 | 6 |
| Urothelial cancer | 0.769 | 8 |

cancer state. This may imply a higher correlation between fluxes in the cancer state for 3 out of 4 cancer types.

**Cancer states require fewer controllers except in lung cancer.** Based on the PMDS model described in the Methods section, we computed the number of reactions necessary to achieve full control of each network in a probabilistic context of flux correlations. We calculated the PMDS for each network using different values of $\Theta$ (threshold probability). The results (Fig. 5) show that the fraction of driver nodes is smaller in all cancer tissues compared to their corresponding healthy states, suggesting that it may be theoretically easier to control these cancer states. This change is consistent with an increase in flux correlations observed in three out of four cancer states. In lung the PMDS fraction is still smaller in the cancer state, even though the healthy state shows higher flux correlations than the cancer state. The size of the largest component is higher in the healthy state for lung, which may explain the observed higher correlations.

To account for possible inaccuracies in the original models, we verified that these results are stable with respect to small

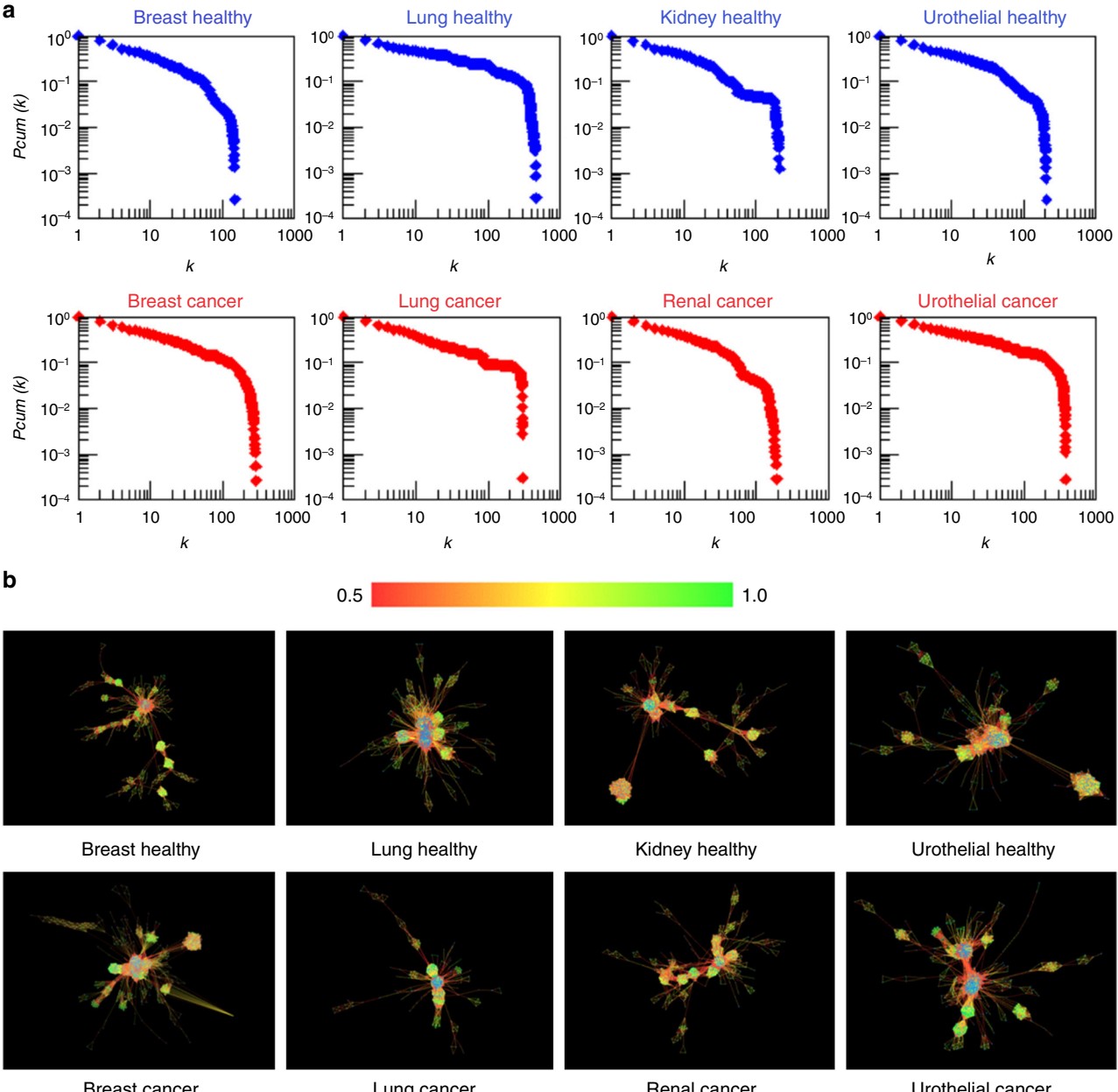

**Fig. 3** Topology of healthy and cancer networks. **a** Cumulative degree distributions of healthy and cancer high flux correlation networks. **b** Visual representation of high flux correlation networks; the edge colour represents the flux correlation ranging from 0.5 (red) to 1.0 (green)

**Table 2 Properties of high flux correlation networks in healthy and cancer states of human tissues**

| Network | Total network size | Size of giant component | Fraction of giant component |
|---|---|---|---|
| Breast healthy | 3729 | 611 | 0.164 |
| Breast cancer | 3741 | 1103 | 0.295 |
| Lung healthy | 3510 | 1091 | 0.311 |
| Lung cancer | 3350 | 598 | 0.179 |
| Kidney healthy | 3986 | 711 | 0.178 |
| Renal cancer | 3444 | 677 | 0.197 |
| Urothelial healthy | 3890 | 660 | 0.170 |
| Urothelial cancer | 3618 | 1078 | 0.298 |

alterations in the topologies of the four healthy and cancer networks. For each tissue type and for each value of $\Theta$, we constructed 10 randomised networks by rewiring 1%, 5% and 10% of the edges, respectively. In order to conserve the topological properties of the networks during randomisation, we applied an edge rewiring algorithm that preserves the degree distribution (see Methods). Even with a moderate perturbation of 10%, the average PMDS fraction only marginally increased and the PMDS fraction in the healthy state always remained larger than in the corresponding cancer state (Supplementary Figs 1, 2 and 3).

**Distribution of controllers in metabolic pathways.** In order to compare the controller and non-controller nodes from a perspective of biological significance, we analysed the distribution of

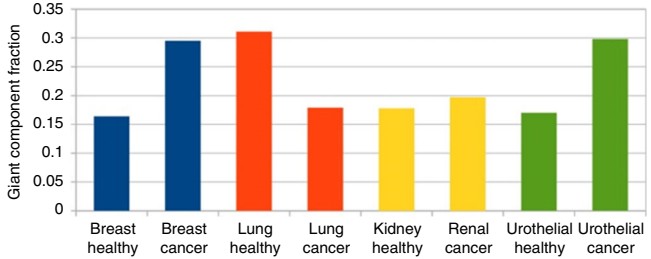

**Fig. 4** Fraction of giant component in healthy and cancer tissues. In these networks, nodes represent reactions and two reactions are connected if the absolute flux correlation between them is higher than 0.5, as described in Methods. The giant component is the largest connected component in each network; the fraction was calculated by dividing the number of nodes in the giant component through the total number of nodes in each network

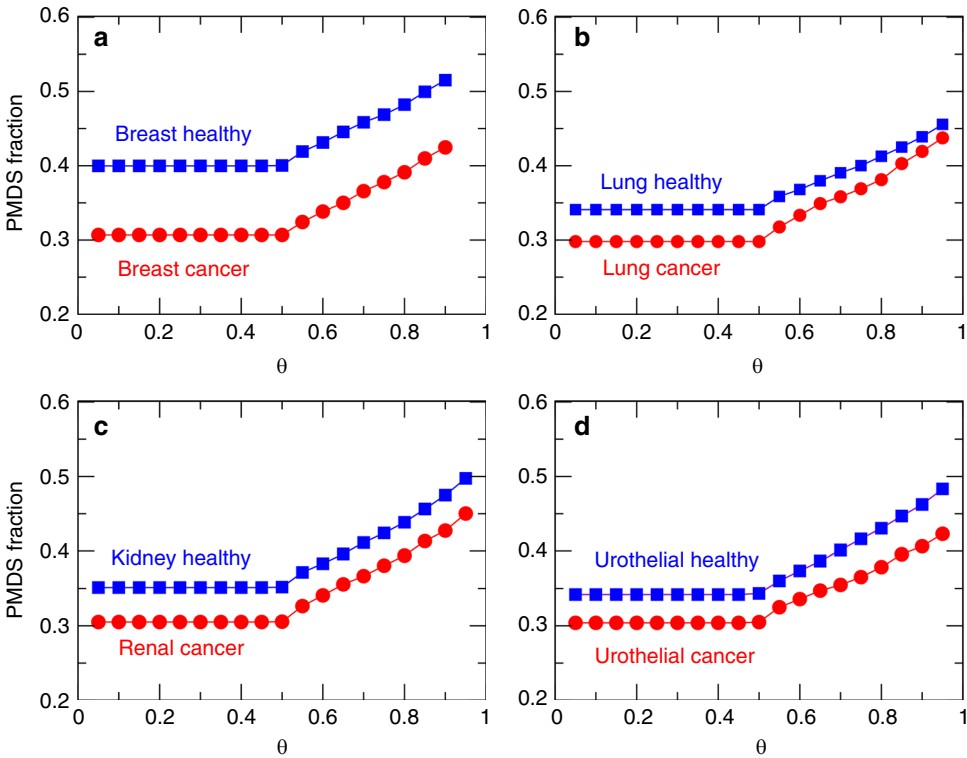

**Fig. 5** Fraction of PMDS in healthy and cancer tissues. **a–d** The PMDS fraction in four healthy (blue) and cancer (red) tissues is displayed for different values of the threshold $\Theta$

PMDS and non-PMDS nodes among metabolic pathways from the KEGG database. Since we are using metabolic models, a comparison to metabolic pathways is more appropriate than for example the Gene Ontology, which contains other types of biological functions not included in the data. We found that some central metabolic pathways are enriched in PMDS nodes across cancer and non-cancer networks: these pathways include glycolysis ($p$-values from 0.009 to 0.06; all $p$-values derived from two-tailed Fisher exact tests) and pyruvate metabolism ($p$-values from 0.003 to 0.06, except renal cancer $p = 0.12$). The citrate cycle is significantly enriched in PMDS nodes in breast and renal tissues ($p$-values from 0.01 to 0.05) but not in urothelial and lung; inositol phosphate metabolism was enriched in healthy breast ($p = 0.03$). Conversely, aminoacyl-tRNA biosynthesis was significantly depleted in PMDS nodes in all cancer and non-cancer networks ($p$-values from 3e−5 to 0.05), as well as N-glycan biosynthesis ($p$-values from 3e−5 to 0.03). Overall, the fact that a higher proportion of controller nodes are found in central

metabolic pathways is consistent with expectations, since these pathways distribute fluxes towards other parts of the metabolic network, and this property is maintained between healthy and cancer cells. Additionally, these results confirm that each cancer type has distinctive characteristics in terms of pathways enriched in PMDS nodes, with lung cancer being more distinct than the three other types.

**Folate cycle subnetwork.** There is growing evidence of relations between metabolic perturbations and cancer. In particular, serine and glycine pathways were found to be associated with oncogenesis[2]. These amino acids feed into the folate cycle, whose outputs feed into the synthesis of nucleotides and phospholipids. To illustrate how control analysis can shed light on changes occurring in metabolic pathways, we show the results obtained on a subnetwork centred on the folate cycle (Fig. 6).

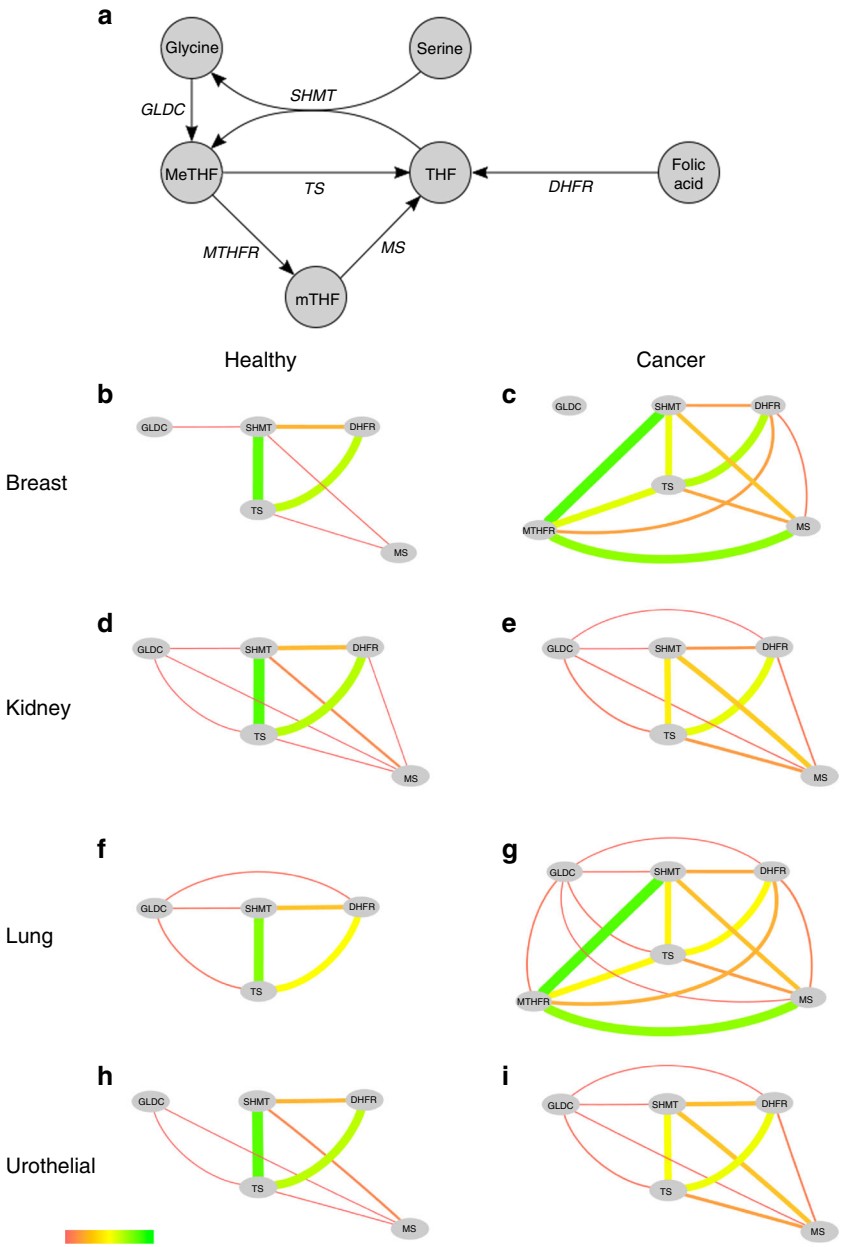

**Fig. 6** Flux correlation changes in the folate cycle. **a** Simplified representation of the folate cycle. In this representation, nodes represent metabolites and edges represent metabolic reactions. **b**–**i** Reaction correlation coefficients in the metabolic system depicted in A for four types of healthy and cancer tissues. In these representations, nodes represent metabolic reactions and the edge colour represents the flux correlation value between the reactions connected by the edge; the colour bar represents the correlations value from red (0.5) to green (1.0). Abbreviations: THF, tetrahydrofolate; MeTHF, 5–10-methylene-tetrahydrofolate; mTHF, 5-methyl-tetrahydrofolate; DHFR, dihydrofolate reductase; GLDC, glycine decarboxylase; MS, methionine synthase; MTHFR, methylene-tetrahydrofolate reductase; SHMT, serine hydroxymethyl transferase; TS, thymidylate synthase

In the healthy states, we observe that SHMT and TS are strongly coupled (Fig. 6b, d, f, h) indicating that the production of THF is strongly coupled to its methylation into MeTHF. In addition, there is a strong positive correlation between TS and DHFR, which means that THF production from folic acid is also coupled to the THF cycle. In the cancer states though, these correlations become weaker (Fig. 6c, e, g, i) and several new connections appear. In breast and lung cancer, MTHFR becomes strongly coupled to SHMT and MS, suggesting that the folate cycle uses mTHF as an intermediate rather than direct transformation of MeTHF to THF. Overall, the cancer states induce more flux interactions between reactions of the THF system. This increases the complexity of the relations and at the same time opens more possibilities to control the fluxes in the system.

## Discussion

Links between metabolism and cancer have been known for a long time, as sustained aerobic glycolysis is well known to occur in cancer cells[15]. How these changes are connected to cell proliferation and accumulation is not well understood though. It was also widely observed that obesity increases cancer risk, which raises questions about the existence of causal links between

increased metabolic activity and tumour progression. Conversely, large animals with low metabolic rate are generally found to have lower incidence of cancer than smaller animals with faster metabolic rate, which strengthens the view that metabolic activity could be a factor favouring cancer, at least as important as mutations[16]. Several oncogenes are known to stimulate metabolic pathways, in particular glucose and glutamine metabolism[17]. Alternative carbon sources such as acetate were found to be better utilised by tumour cells than normal tissue[18]. Metabolic flux distributions in tumour cells were observed to correlate with increased lactate production[19] and to maintain levels of NADPH, allowing cells to better resist to oxidative stress[20,21].

Our results show that, in the four tissue types for which comparative models of healthy and cancer metabolic networks are available, namely breast, lung, kidney and urothelial, controllability in cancer metabolism is easier than that in healthy metabolism. A generic interpretation of this result can be found by combining knowledge from Fig. 3. The network visualization shows significant flux correlation changes. However, the cumulative degree distribution shows that, in spite that many reactions change flux correlations, the global statistical degree pattern of the network does not change and still follows a power-law for the degree of nodes. This is important technically because in general the PMDS can be computed faster in networks that have a power-law distribution, i.e. scale-free networks. However, the new flux distribution in cancer state seems to be more highly correlated (lower failure probabilities) as shown in Fig. 2, which makes cancer states easier to control from a PMDS view point since key flux routes have lower probability of failure. Indeed, these results are based on comparative analyses of metabolic reconstructions of cancer and healthy tissues and remain dependent on the quality of the models used. Nevertheless, we verified that these results are preserved under moderate alterations of the networks, and as the quality of metabolic models increases and new models become available for other tissue types, these properties can be further tested in the future. It is worth mentioning that other types of analyses not related to controllability may be able to identify differences between cancer and healthy metabolism. For example null-space analysis can characterise properties of genome-scale metabolic networks based on stoichiometry alone[13] and other types of constraints such as thermodynamics may be taken into consideration.

Nevertheless, specific differences in metabolic flux correlations were observed in lung cancer. These differences were also reflected in the number of reactions assembled in the main connected component of the network. Heterogeneity in metabolic pathway activity has been reported before, not only between different types of cancers[22] but also between different types of lung tumours[23,24]. In spite of the heterogeneous metabolic flux response observed in cancers and in particular a large fluctuation of active correlations (i.e. number of links) among reactions in lung cancer, the number of necessary reactions to be controlled in cancer does not largely change. More importantly, the metabolic flux space in cancer remains easier to control than that in healthy state.

The number of active correlations in lung cancer is twice as large as that number in breast cancer. Counterintuitively, the PMDS size of both lung and breast cancer is lower than that in the corresponding healthy state. This indicates that the size of the reaction control backbone of both lung and breast cancer is very similar. The increase in correlations observed in lung cancer, consistent with the already reported variability on cancer metabolism, might only perform peripheral metabolic functions without critical control roles. Glutamine metabolism is altered in many types of cancer cells, but the consumption of glutamine by lung cancer cells is higher than in other cancer types[25].

The glutamine metabolic pathway is tightly interconnected with the mTOR signalling pathway, which promotes cell survival and is activated by glutamine efflux; this particular feature is being investigated for potential therapeutic applications[26]. The tyrosine kinase epidermal growth factor receptor (EGFR) is frequently mutated in non-small cell lung cancer and has strong interrelations to several metabolic pathways. It was shown that EGFR signalling promotes not only glucose consumption and lactate production, but also de novo pyrimidine synthesis, therefore it plays a major regulatory role on global metabolism[27]. Different KRAS mutations are also found in lung cancer, which do not have the same cellular activity. It was shown that they affect different metabolic pathways, with distinguishing effects in particular in the glutaminolysis and glutathione pathways[28]. These examples show that, in addition to strong metabolic effects shared with other types cancers such as aerobic glycolysis, lung cancers can also be characterised by original relations with metabolic pathways. The full extent of these relations is still poorly understood but is an active area of research towards new therapies[25].

## Methods

**Computation of reaction correlation coefficients**. The computation starts by constructing the stoichiometric matrix of the network, $\mathbf{S}$, whose elements are the stoichiometric coefficients of each metabolite in each reaction. The kernel matrix $\mathbf{K}$ is defined by vectors constituting a basis of the null-space of $\mathbf{S}$, such that $\mathbf{S}\,\mathbf{K}^T = 0$.

In most cases, $\mathbf{K}$ is not unique. However, if the vectors of $\mathbf{K}$ form an orthonormal basis, which means $\mathbf{K}\,\mathbf{K}^T = \mathbf{I}$, then the angles $\theta_{ij}$ between the row vectors of $\mathbf{K}$ are invariant. The reaction correlation coefficient $\phi_{ij}$ is defined as the cosine of $\vartheta_{ij}$:

$$\phi_{ij} = \frac{\boldsymbol{k}_i \boldsymbol{k}_j^T}{\sqrt{\left(\boldsymbol{k}_i \boldsymbol{k}_i^T\right)\left(\boldsymbol{k}_j \boldsymbol{k}_j^T\right)}} = \cos(\vartheta_{ij}) \qquad (1)$$

where $\boldsymbol{k}_i$ and $\boldsymbol{k}_j$ are the row vectors of $\mathbf{K}$ corresponding to reactions $i$ and $j$ respectively. As demonstrated in Poolman et al.[13], $\phi_{ij}$ represents the Pearson correlation coefficient between the fluxes carried by reactions $i$ and $j$ over all possible steady states of the system. The mathematical demonstration is based on introducing a random matrix $\mathbf{R}$ that contains all possible steady states, $s$ being the number of steady states which can be arbitrarily large. Then the Pearson correlation $r$ of this distribution is defined, and when $s$ tends towards infinite it is shown that $r$ tends towards a product of $\boldsymbol{k}$ vectors, which represents the cosine of the angle.

We calculated $\mathbf{K}$ using the *null* function in Matlab and verified that it meets the orthonormality condition for each metabolic network. After obtaining $\phi_{ij}$ for each pair of reactions, the probability of failure between reactions $i$ and $j$ was defined as $\rho_{ij} = 1 - \mathrm{abs}(\phi_{ij})$.

**Probabilistic control model**. Natural and engineered complex networks are composed of thousands of nodes and tens of thousands of links. These links representing regulatory interactions or transmission lines suffer from probabilistic failures. The flux correlation coefficient allows us to define the probability of failure between reactions $i$ and $j$ ($\rho_{ij}$) and integrate it into a probabilistic control model. We want each node (reaction) to be covered by multiple nodes in MDS so the probability that at least one edge (incoming flux) is active is at least $\Theta$. This problem can be formulated as a probabilistic minimum dominated (PMDS) as follows. Let $S$ be a dominating set, then we require $S$ to satisfy:

$$\left(1 - \prod_{i \in S} \rho_{ij}\right) \geq \Theta, \; \forall j \in V \qquad (2)$$

which can be rewritten as:

$$\sum_{i \in S} -\ln\left(\rho_{ij}\right) \geq -\ln(1 - \Theta) \qquad (3)$$

where $\rho_{ij}$ indicates the probability of failure between reactions $(i, j)$.

The standard MDS problem is formalized by the following integer linear programming (ILP) problem:

*Minimise* $\displaystyle\sum_{i \in V} x_i$

*subject to*:

$$x_i + \sum_{(j,i) \in E} x_j \geq 1, \qquad (4)$$

$$x_i \in \{0, 1\}, \; \forall i \in V,$$

where an MDS is obtained by the set $\{x|x_i = 1\}$, $V$ indicates the set of reaction nodes in the network and $E$ denotes the set of undirected edges between reaction nodes. Then, inserting the probabilistic condition shown in Eqs 2 into 3 leads to the PMDS formalized as the following ILP:

$$\text{Minimise} \sum_{i \in V} x_i,$$

$$\text{subject to: } x_j \geq 1, \; \forall j \in V \text{ such that } \deg(j) = 0,$$

$$-\ln(1 - \Theta)x_j + \sum_{(i,j)\in E}\left(-\ln(\rho_{ij})x_i\right) \geq -\ln(1 - \Theta), \; \forall j \in V \text{ such that } \deg(j) > 0,$$

$$(5)$$

$$x_i \in \{0, 1\}, \; \forall v \in V$$

where $deg(j)$ denotes the degree of reaction node $j$. In the above expression, the first term $-\ln(1 - \Theta)x_j$ is added because if node $j$ is included in the PMDS, the inequality needs to be hold. In the implementation of the ILP-based method, a small number ($10^{-6}$) is added to $\rho_{ij}$ to avoid infinity occurrence at $\rho_{ij} = 0$. The ILP problem was solved using the IBM ILOG CPLEX Optimiser package version 12.0.

**Construction of cancer networks.** Genome-scale models representing human metabolic pathways in healthy and cancer states were compiled from Basler et al.[5] and Gatto et al.[29] Supplementary data. Then, flux correlation networks were constructed by computing the reaction correlation coefficients. To create healthy and cancer networks a threshold of 0.5 was used, which means that absolute correlations smaller than 0.5 were deleted, or conversely that failure probabilities higher than 0.5 were deleted; in the resulting networks, nodes represent reactions and two reactions are connected if their correlation is higher than 0.5. This means that we are considering the high flux correlation networks in our analysis.

**Construction of randomised networks.** For each tissue type and for each value of $\Theta$, we constructed 10 randomised networks by rewiring 1%, 5% and 10% of the edges, respectively. We used the rewire function from the R igraph package, together with the keeping_degseq function that preserves the original network's degree distribution, in order to conserve the topological properties of the networks.

**Pathway enrichment analysis.** We analysed the distribution of PMDS and non-PMDS nodes across all metabolic pathways of the KEGG database in order to determine whether some pathways are significantly enriched or depleted in controller nodes. Enzyme Commission numbers associated to reactions were extracted from the raw metabolic models, then were mapped to PMDS and non-PMDS node lists in the same conditions as described above. These node lists were searched against KEGG pathways using the KEGG Mapper tool[30] in order to obtain the number of PMDS and non-PMDS nodes in each pathway, then two-tailed Fisher exact tests were conducted using R in order to determine significant enrichment or depletion.

**Reporting summary.** Further information on research design is available in the Nature Research Reporting Summary linked to this article.

## Data availability

The models used in this study are available from https://doi.org/10.1101/gr.202648.115, Supplemental Material section. Data that support the tables and figures of this study are available from the corresponding authors upon request.

## Code availability

Custom code used in this study is available from the corresponding authors upon request.

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

## Acknowledgements

J.C.N. thanks the Royal Society for an International Exchange grant (IE160248). J.C.N. was partially supported by JSPS KAKENHI Grant Number 18K11535. T.A. was partially supported by JSPS KAKENHI Grant Number 18H04113. This research was also supported in part by Research Collaboration Projects of the Institute for Chemical Research, Kyoto University.

## Author contributions

J.M.S. and J.C.N. designed the study, performed the analysis and wrote the manuscript. J.M.S., H.O. and J.C.N. analysed the data. T.A. contributed to the theoretical discussion and data analysis interpretation. All authors read and approved the final manuscript.

## Additional information

**Competing interests:** The authors declare no competing interests.

