## [Peer Review File · Nature Communications]

Reviewers' comments:

Reviewer #1 (Remarks to the Author):

Schwartz et al used a Probabilistic Minimum Dominating Set (PMDS) model to identify a minimum set of nodes that act as driver nodes and control the entire network using previously reconstructed healthy and cancer models. The authors proposed a new definition of flux coupling that is measured by a continuous value characterising the relations between reaction fluxes over all feasible steady states and combined it with an advanced probabilistic control theory method to analyse healthy-cancer transitions in metabolic networks. Their analysis showed that controlling metabolic fluxes in cancer requires fewer controllers than in the healthy state, possibly because the flux correlations substantially increase in cancer states.

The theoretical work have been performed carefully by the authors. The authors used an existing methodology and models to analyse the differences between healthy and cancer tissues. The study relies on the certain differences between the fluxes. Considering that the authors have not used the measured uptake and secretion rates, the predicted fluxes can only provide information based on the topology of the models. One limitation of the genome-scale metabolic models is the large solution space. In other words, the fluxes can vary just based on the topology of the network. Considering that these models are generated based on transcriptomics data, the topology should not be used for flux balance analysis without any measured secretion and excretion rates.

In order to consider the paper for publication in Nat. Com, I recommend the authors validate the predicted fluxes in cell lines isolated from healthy tissues and cancers. Otherwise, the study is theoretical and suitable for a specialized journal.

Reviewer #2 (Remarks to the Author):

This manuscript addresses the problem of using probabilistic control theory method to analyse healthy-cancer transitions in genome-scale metabolic networks.

In this work, the authors first compiled metabolic networks corresponding to four healthy and cancer tissues, including breast, lung, kidney and urothelial cancer. The nodes in the networks correspond to metabolic reactions and the weighted edges correspond to the probabilistic flux exchanged among reactions.

Compared with previous works, this work measured flux coupling by a continuous value characterising the relations between reaction fluxes over all feasible steady states, instead of binary values.

Then, a Probabilistic Minimum Dominating Set (PMDS) model was used to identify a minimum set of nodes that act as driver nodes and control the entire networks.

The manuscript has an interesting starting point, i.e., the first time to connect control theory with probabilistic flux correlations by PMDS, which was used to analyse the healthy-cancer transition from the metabolic flux changes perspective.

My problems are as follows:

1) In abstract,controlling metabolic fluxes in cancer requires fewer controllers than in the healthy state, possibly because the flux correlations substantially increase in cancer states... This sentence is too simple to show the findings in this paper.

2) In Table 1, please show the biological significances of different connected components in healthy and cancer networks.

3) In the top of Fig. 3, it is harder to read. Please improve the quality of the figures.

4) The results of lung are different from that of breast, kidney and so on. More detailed information should be provided in the revised version.

5) In Fig. 6 D, E, combined with 4). More descriptions should be added.

6) Compare the driver nodes or called controllers in this paper with non-controllers from a perspective of biological significances.

General comments:

This manuscript is very interesting, more biological significances should be provided to show the findings.

Manuscript NCOMMS-18-17471 – Response to Reviewers' comments

We thank both reviewers for their positive assessment of our manuscript. We provide a point-by-point response to their comments below and have revised the manuscript accordingly. New and revised sections are highlighted in red in the updated manuscript.

Reviewer #1

Schwartz et al used a Probabilistic Minimum Dominating Set (PMDS) model to identify a minimum set of nodes that act as driver nodes and control the entire network using previously reconstructed healthy and cancer models. The authors proposed a new definition of flux coupling that is measured by a continuous value characterising the relations between reaction fluxes over all feasible steady states and combined it with an advanced probabilistic control theory method to analyse healthy-cancer transitions in metabolic networks. Their analysis showed that controlling metabolic fluxes in cancer requires fewer controllers than in the healthy state, possibly because the flux correlations substantially increase in cancer states.

The theoretical work have been performed carefully by the authors. The authors used an existing methodology and models to analyse the differences between healthy and cancer tissues. The study relies on the certain differences between the fluxes. Considering that the authors have not used the measured uptake and secretion rates, the predicted fluxes can only provide information based on the topology of the models. One limitation of the genome-scale metabolic models is the large solution space. In other words, the fluxes can vary just based on the topology of the network. Considering that these models are generated based on transcriptomics data, the topology should not be used for flux balance analysis without any measured secretion and excretion rates.

In order to consider the paper for publication in Nat. Com, I recommend the authors validate the predicted fluxes in cell lines isolated from healthy tissues and cancers. Otherwise, the study is theoretical and suitable for a specialized journal.

We thank the reviewer for noting the novelty and importance of our work. We believe there is a misunderstanding though when the reviewer is asking to “validate the predicted fluxes in cell lines”: our paper does not present any flux predictions, but we compute “flux correlations” which represent the average correlation between fluxes of reaction pairs over all feasible steady states. This constitutes a major innovation of this work since previous studies about “flux coupling” were based on a very specific subset of flux distributions, while our method takes into account the full steady-state solution space. Even if it were possible to measure uptake and secretion rates of some metabolites in different cell lines, these experiments could only provide a limited subset of flux values over a limited subset of conditions. To provide an experimental validation of flux correlations would require measuring genome-scale reaction fluxes (not only external but also internal) over a comprehensive series of conditions, which is not currently technically feasible even by the most advanced methods.

We also note that the reviewer mentioned the “large solution space” to be a “limitation” of genome-scale metabolic models, but our method precisely addresses that limitation since it encompasses the full solution space. The mathematical demonstration of this property was

published in Poolman et al. (*J Theor Biol* **249**, 691-705, 2007), which is referenced in our paper.

Instead of conducting biological experiments, we have carried out a systematic analysis of biological functions of controllers and non-controllers in the different networks (page 13), and have expanded the discussion to provide more information about differences between lung and other types of cancer (pages 16-17).

Reviewer #2

This manuscript addresses the problem of using probabilistic control theory method to analyse healthy-cancer transitions in genome-scale metabolic networks.

In this work, the authors first compiled metabolic networks corresponding to four healthy and cancer tissues, including breast, lung, kidney and urothelial cancer. The nodes in the networks correspond to metabolic reactions and the weighted edges correspond to the probabilistic flux exchanged among reactions.

Compared with previous works, this work measured flux coupling by a continuous value characterising the relations between reaction fluxes over all feasible steady states, instead of binary values.

Then, a Probabilistic Minimum Dominating Set (PMDS) model was used to identify a minimum set of nodes that act as driver nodes and control the entire networks.

The manuscript has an interesting starting point, i.e., the first time to connect control theory with probabilistic flux correlations by PMDS, which was used to analyse the healthy-cancer transition from the metabolic flux changes perspective.

My problems are as follows:

1) In abstract,controlling metabolic fluxes in cancer requires fewer controllers than in the healthy state, possibly because the flux correlations substantially increase in cancer states... This sentence is too simple to show the findings in this paper.

We have expanded the abstract by adding more details about our results as follows:

“The combination of control theory with flux correlation analysis showed that flux correlations substantially increase in cancer states of breast, kidney and urothelial tissues, but not in lung. No change in the network topology between healthy and cancer networks was observed, but PMDS analysis showed that cancer states require fewer controllers than their corresponding healthy states. Together these results indicate that cancer metabolism is characterised by more streamlined flux distributions, which may be focused towards a reduced set of objectives such as growth and controlled by a reduced set of regulatory elements.”

2) In Table 1, please show the biological significances of different connected components in healthy and cancer networks.

Table 1 refers to the full metabolic network therefore there is no particular biological significance or enriched function found in these components. The main object of Table 1 is to show that cancer and healthy networks have similar topological properties, indicating that topology alone cannot explain the differences found in flux correlations, hence justifying the need for a new type of analysis such as PMDS.

3) In the top of Fig. 3, it is harder to read. Please improve the quality of the figures.

We have increased the size of labels in the top part of Figure 3.

4) The results of lung are different from that of breast, kidney and so on. More detailed information should be provided in the revised version.

We have expanded the discussion as follows to provide more information about differences between lung and other types of cancer, and added several new references (pages 16-17).

“Glutamine metabolism is altered in many types of cancer cells, but the consumption of glutamine by lung cancer cells is higher than in other cancer types (Mohamed et al., 2014). The glutamine metabolic pathway is tightly interconnected with the mTOR signalling pathway, which promotes cell survival and is activated by glutamine efflux; this particular feature is being investigated for potential therapeutic applications (Fumarola et al., 2014). The tyrosine kinase epidermal growth factor receptor (EGFR) is frequently mutated in non-small cell lung cancer and has strong interrelations to several metabolic pathways. It was shown that EGFR signalling promotes not only glucose consumption and lactate production, but also *de novo* pyrimidine synthesis, therefore it plays a major regulatory role on global metabolism (Makinoshima et al., 2014). Different KRAS mutations are also found in lung cancer, which do not have the same cellular activity. It was shown that they affect different metabolic pathways, with distinguishing effects in particular in the glutaminolysis and glutathione pathways (Brunelli et al, 2014). These examples show that, in addition to strong metabolic effects shared with other types cancers such as aerobic glycolysis, lung cancers can also be characterised by original relations with metabolic pathways. The full extent of these relations is still poorly understood but is an active area of research towards new therapies (Mohamed et al., 2014).”

5) In Fig. 6 D, E, combined with 4). More descriptions should be added.

We have expanded the legend descriptions of figures 4 and 6.

6) Compare the driver nodes or called controllers in this paper with non-controllers from a perspective of biological significances.

We carried out a systematic analysis of biological functions or controllers and non-controllers in the different networks. We added the results of this analysis to a new section of the Results in our manuscript (page 13). We also added a new section in the Methods explaining how these enrichments were computed (page 7).

“In order to compare the controller and non-controller nodes from a perspective of biological significance, we analysed the distribution of MDS and non-MDS nodes among metabolic pathways from the KEGG database. Since we are using metabolic models, a comparison to metabolic pathways is more appropriate than for example the Gene Ontology, which contains other types of biological functions not included in the data. We found that some central metabolic pathways are consistently enriched in MDS nodes across all cancer and non-cancer networks: these pathways include glycolysis (p-values from 0.003 to 0.03) and pyruvate metabolism (p-values from 0.0002 to 0.03). The citrate cycle is significantly enriched in MDS nodes in breast, urothelial and renal tissues (p-values from 0.0004 to 0.05) but not in lung ($p > 0.05$). A few other pathways were enriched in MDS nodes only in specific conditions: for example purine metabolism ($p = 0.01$) and glyoxylate and dicarboxylate metabolism ($p = 0.002$) were enriched in breast cancer; inositol phosphate metabolism was enriched in healthy breast ($p = 0.007$); pyrimidine metabolism was enriched in lung cancer ($p = 0.005$) and urothelial cancer ($p = 0.03$). Conversely, aminoacyl-tRNA biosynthesis was significantly depleted in MDS nodes in all cancer and non-cancer networks (p-values from 0.0004 to 0.003), and N-glycan biosynthesis was significantly depleted in MDS nodes in breast, urothelial, renal and healthy lung (p-values from 0.0005 to 0.04), but not in lung cancer. Overall, the fact that a higher proportion of controller nodes are found in central metabolic pathways is consistent with expectations, since these pathways distribute fluxes towards other parts of the metabolic network, and this property is maintained between healthy and cancer cells. Additionally, these results confirm that each cancer type has distinctive characteristics in terms of pathways enriched in MDS nodes, with lung cancer being more distinct than the three other types.”

General comments:

This manuscript is very interesting, more biological significances should be provided to show the findings.

We thank the reviewer for this supportive comment.

Reviewers' comments:

Reviewer #1 (Remarks to the Author):

As I clearly mentioned in my previous review, the fluxes can vary just based on the topology of the network. Considering that these models are generated based on transcriptomics data, the topology should not be used for flux balance analysis without any measured secretion and excretion rates.

In order to consider the paper for publication in Nat. Com, the authors should validate the predicted fluxes in cell lines. Otherwise, the study is theoretical and suitable for a specialized journal.

I would not support the publication of such method without any experimental data. Me and other reviewer have provided extremely supportive comments but the authors did not perform any experiments or provide experimental evidence for the superiority of their method.

There are large number of flux and labelling studies available in the literature from cancer studies and the authors could use such data to provide experimental evidence. Without such data it is impossible to evaluate such computational method,

I am really sorry but I can not support the publication of this paper in Nat. Com.

Reviewer #2 (Remarks to the Author):

The authors have carefully addressed my problems, and I have no more comments on this version.

This paper could be accepted for publication in NComms.

Reviewer #3 (Remarks to the Author):

The work by Schwartz et al reports an interesting analysis that is shown to allow one to differentiate between healthy and cancerous tissue based on topological information only. The method is innovative and I believe of interest.

In his/her second response Reviewer #1 makes clear that he/she is aware that the authors do not predict fluxes. However, he/she stresses that the topology – as captured in the stoichiometric matrix S – is not reliable and requires verification. I agree with this statement, which has recently been highlighted as well (doi:10.1016/j.cels.2017.01.010). Whether or not flux data are especially useful (with respect to validating a genome-scale metabolic network), may be a matter of discussion.

I do want to add one concern of my own: The authors say that a key advantage of their method is that it allows one to consider the totality of the available solution space as characterized by the null-space of the stoichiometric matrix. However, the solution space of the equation $Sv = 0$ is unbounded and most solutions are biologically irrelevant, as for instance thermodynamic constraints [reaction (irr-)reversibilities] or allocation and capacity constraints are not considered. Therefore, it remains open whether or not the (sub)space of biologically relevant solutions, shows the same behavior as the solution in the null-space. Clearly, flux data could be used to identify the relevant solution space. The statements of reviewer #1 could be seen as hinting at those issues as well.

While I don't think experiments are essential, I suggest checking the stability of the predictions with respect to (randomly selected, small) alterations in the topologies of the four networks. I am aware that this might not be so easy, as any changes of the network need to keep the scale-freeness of the network, and – in case of additions to the network – these reactions need to be flux carrying as well.

In addition, the claims in the manuscript should be carefully phrased. For instance, "Cancer states require fewer reactions to be controlled except in lung cancer". To me this sentence is extremely general, which is not supported by the data. First, the statement is based on an analysis of mathematical models of cancer. The quality of these models is debatable (see reviewer #1). Second, four points of validation do not allow to generalize to the totality of all cancer states without states in lung cancer. It may very well be that lung cancer is the typical state and the three other cancer types are the exception to the rule. I acknowledge that in the presentation of their results the authors formulate carefully. However, in the discussion section this reservation is a bit lost. The authors should highlight the theoretical nature of their results and clearly point at gaps and possible shortcomings in their analysis.

Manuscript NCOMMS-18-17471 – Response to Reviewers' comments

We thank the reviewers for their positive assessment of our manuscript. We provide a point-by-point response to their comments below and have revised the manuscript accordingly. The revised sections are highlighted in red in the updated manuscript.

Reviewer #3

While I don't think experiments are essential, I suggest checking the stability of the predictions with respect to (randomly selected, small) alterations in the topologies of the four networks. I am aware that this might not be so easy, as any changes of the network need to keep the scale-free-ness of the network, and – in case of additions to the network – these reactions need to be flux carrying as well.

We thank the reviewer for this helpful suggestion. In order to check the stability of our predictions, we constructed a series of randomised networks using different levels of perturbations, and we compared the PMDS fractions in the randomised networks to those of the unperturbed networks. It is important indeed to conserve the topological properties during randomisation to avoid introducing biases, therefore we applied an edge rewiring algorithm that conserves the degree distribution. We have added a new section in the Methods to describe the randomisation procedure (page 7) and another section in the Results to present the outcomes of this new analysis (page 12). It turned out that the PMDS fraction increases in the randomised networks, but this increase remains small even for moderate (10%) alterations in the networks. The PMDS fraction always remains higher in the cancer network compared to the corresponding healthy tissue network, confirming that our predictions are stable. We present the new data in three new Supplementary Figures which are referred to in the main text.

In addition, the claims in the manuscript should be carefully phrased. For instance, "Cancer states require fewer reactions to be controlled except in lung cancer". To me this sentence is extremely general, which is not supported by the data. First, the statement is based on an analysis of mathematical models of cancer. The quality of these models is debatable (see reviewer #1). Second, four points of validation do not allow to generalize to the totality of all cancer states without states in lung cancer. It may very well be that lung cancer is the typical state and the three other cancer types are the exception to the rule. I acknowledge that in the presentation of their results the authors formulate carefully. However, in the discussion section this reservation is a bit lost. The authors should highlight the theoretical nature of their results and clearly point at gaps and possible shortcomings in their analysis.

In order to phrase our claims more carefully, we modified the second paragraph of the Discussion (pages 16-17) to make it clear that these results are based on the analysis of four specific cancer and healthy tissue models and that they are dependent on the quality of these models.

REVIEWERS' COMMENTS:

Reviewer #3 (Remarks to the Author):

My suggestions have been appropriately addressed and the documented stability is reassuring.

However, in my original review I also raised the issue that the authors characterize cancer in terms of the null-space which disregards thermodynamic or allocation or other constraints. Unfortunately the authors did not comment on this issues. (Admittedly I did not raise a specific question to that point other than saying it remains open.) Personally, I encourage the authors to address this issue in a brief statement in the discussion, especially with respect to possible further work.

Manuscript NCOMMS-18-17471 – Response to Reviewers' comments

We thank the reviewer for his/her positive assessment of our manuscript. We address the remaining comment below.

Reviewer #3

However, in my original review I also raised the issue that the authors characterize cancer in terms of the null-space which disregards thermodynamic or allocation or other constraints. Unfortunately the authors did not comment on this issues. (Admittedly I did not raise a specific question to that point other than saying it remains open.) Personally, I encourage the authors to address this issue in a brief statement in the discussion, especially with respect to possible further work.

Indeed we did not analyse the null-spaces of these metabolic models since in our view this would be a different type of analysis not closely related to controllability, which is the main subject of this article. However, we understand the importance of considering constraints such as thermodynamics and other types. We have now added a statement in the discussion (page 7) mentioning these points for future work.